# Novel Technique for Design and Manufacture of Alternating Gradient Composite Structure of Aluminum Alloys Using Solid State Additive Manufacturing Technique

**DOI:** 10.3390/ma15207369

**Published:** 2022-10-21

**Authors:** Hari Venkit, Senthil Kumaran Selvaraj

**Affiliations:** Department of Manufacturing Engineering, School of Mechanical Engineering, Vellore Institute of Technology (VIT), Vellore 632014, India

**Keywords:** additive manufacturing, friction stir lap welding, friction stir additive manufacturing, mechanical property, microstructure and fractography

## Abstract

This work analyzes a novel solid-state manufacturing approach of a friction stir additive manufacturing (FSAM) technique for fabricating multiple layers of alternating gradient composite structure using alternate layers of AA6061-T6 and AA7075-T6 aluminum alloys of 3 mm thickness. The evolution of the microstructure along the build direction and its impact on the tensile and microhardness properties were examined using optical microscopy, tensile tests, and Vickers microhardness tests. Nonuniform microstructures were detected along the build direction, and it was concluded that the most productive part of the construction was the nugget zone, which had fine equiaxed grains. It was identified that the grain sizes and precipitate sizes were affected by the varying thermal cycles created by the multiple passes of the tool. These events were identified as the primary reasons for the increase in strength and hardness of the FSAM build from the lower layer to the upper layer. In the final FSAM build the maximum hardness value was obtained as 182.3 HV and the ultimate tensile strength (UTS) was 420 MPa both of which were identified at the topmost layer. Moreover, the postmortem of the fractured samples revealed that the cause of failure was a combination of both ductile and brittle fractures. The findings of this study suggest that the FSAM approach may be used to fabricate large structures that are free of defects having expected mechanical characteristics and hence the newly fabricated composite can be used as a suitable substitute for the conventional AA6061 material applied in automobile components for its improved performance.

## 1. Introduction

Various industrial revolutions have sparked the creation of new manufacturing methods throughout the years. The fourth industrial revolution is primarily driven by advances in automation, robotics, and digital fabrication. Additive manufacturing (AM), the most recent development in the fourth industrial revolution, is a cutting-edge technology that allows for the rapid development of complex-shaped objects and is presently changing all manufacturing initiatives [1]. In contrast to subtractive manufacturing methods such as machining, the simplest definition of additive manufacturing is “the technique of combining materials layer by layer to produce things using three-dimensional (3D) model data” [2]. AM employs simple materials such as wires, powders, or blocks to directly manufacture parts of the desired final size. As a result, it provides several benefits, including facilitating more creative designs, lowering the consumption of raw materials to the bare minimum, reducing the time for processing, and thus minimizing the manufacturing cost. This unique technology has lately been widely developed using principles of sophisticated welding and joining techniques in opposition to traditional production processes such as casting, powder technology, and metal forming [3].

Even though AM has risen to prominence due to significant technological breakthroughs, many significant hurdles still remain. Most metal-based AM techniques are limited to procedures based on melting metals with lasers or electron beams and shaped metal deposition, which are all fusion-based additive manufacturing techniques [4]. These techniques have shown that they can be used to make a multi-layered part with complicated shapes and have developed progressively with time. However, these processes still have certain restrictions due to the nature of fusion occurring during these techniques. As a result of the solidification and melting occurring during these processes, various metallurgical defects such as porosity and high crack sensitivity are invariably produced in parts manufactured by these techniques, particularly in alloys such as aluminum [5], titanium [6], and magnesium [7]. To circumvent these defects, various solid-state additive manufacturing technologies such as ultrasonic additive manufacturing (UAM) [8], cold spray additive manufacturing (CSAM) [9], friction stir additive manufacturing (FSAM) [10], etc., were conceived. According to the ASTM categorization system, the aforementioned solid-state techniques are considered subsets of the sheet lamination approach.

FSAM (Figure 1a) is reckoned as an evolving metal additive manufacturing technique where its operating principle is comparable to friction stir lap welding (FSLW). This way of manufacturing is called beamless additive manufacturing, as the raw materials do not melt during the process [11,12]. Multiple layers of metal sheets or plates are joined together by the frictional heat produced by the contact of raw materials and the rotating tool [13]. In this technique, heat generated due to friction is much lower than the temperature at which the base metal melts. Consequently, the microstructure and thermal distortion of the raw material after manufacturing are kept to a minimum [14]. Comparing FSAM to other solid-state techniques, the primary advantages of FSAM are the formation of strong diffused bonds between layers and a greater deposition rate [15]. In this AM technique, multiple layers of dissimilar or similar materials are joined by stacking the plates one over the other, and the joining of these plates is carried out using the FSLW technique until the required height is obtained (Figure 1b) [16]. FSAM is superior to fusion-based AM techniques since it is accomplished in the solid state without melting the metals, thus preventing any solidification defects in the fabricated components and undesirable phase transition in metallurgy. Various studies reveal that the final structure fabricated by the FSAM technique has refined and equiaxed grains, especially in the nugget zone (NZ), with minimal deformations and minimal residual stresses [17].

Airbus was the first to come up with the idea of joining metals using the FSAM technique in 2006. Since then, only a few published research investigations have been made on the FSAM build’s microstructure and monotonic characteristics due to its early stages of development. Palanivel et al. [18] conducted a study on FSAM where 1.7 mm thick WE43 Mg alloy plates were stacked one over the other and joined using the FSAM technique until a final height of 5.6 mm was achieved. They discovered that the stir zone of the FSAM sample had higher ductility, ultimate tensile strength, and yield strength compared with the parent metals despite the presence of volumetric flaws. They also identified complex microstructures in different layers with various degrees of refinement, equiaxed grains, and banding regions and concluded that the varying heat input produced during the FSAM technique was the reason for the complex microstructural evolution. Mao et al. [19] did a similar investigation concentrating on the FSAM of aluminum alloy AA7075. In that study, 5 mm thick plates of AA7075 were joined using the FSAM technique until a building height of 42 mm was achieved. Studies of various mechanical properties revealed that the microhardness and tensile strength decreased from the upper layer to the lower layer of the construction. The friction stir plasticizing of materials created perfect solid-state bonds between the different layers, which made the grain structure of the layered aluminum structure much finer and stronger. Static annealing coarsened the grains and precipitates at the bottom, while the grain size at the top was smaller compared to the middle region. However, the overall elongation of the FSAM AA7075 built was reduced in this investigation. Palanivel et al. [20] also fabricated multi-layers of AA5083-O using the FSAM technique and identified that the strength and microhardness increased as the build height went up. In an interesting work by Zhao et al. [21], multiple layers of 2 mm thick plates of Al-Li alloy were joined over a base plate of AA6061-T6 Al alloy. Hardness profile, elongation, and tensile strength were found to be inhomogeneous along the build height. The presence of oxides at the joint interface, and inadequate stirring of the tool, resulted in the weak-bonding defect in the FSAM part. In a recent study by Roodgari et al. [22], two layers of St52 steel composite structure were fabricated by applying the friction-surfacing route in FSAM technology. Characterization studies were performed on the fabricated composite material, and the bonding between the laminated layers was analyzed. Another exciting work by Changshu He et al. [23] involved the fabrication of multi-layers of 7N01 aluminum alloy using the FSAM technique. A final height of 42 mm was achieved in this process. Due to the existence of defects such as kissing bonds at the interface, they noticed a drop in microhardness as well as in the ultimate tensile strength. Ying et al. [24] developed a multi-layered structure through the use of underwater FSAM technology. This structure was found to have maximal strength in the construct region driven by both the shoulder and pin of the tool. They identified that problems related to local softening could be reduced using the underwater FSAM technique. Very recently, Wlodarski et al. [25] applied the route of multipass FSLW to fabricate a seven-layered AZ31 Mg alloy. The multipass technique was employed to attain a wider weld nugget zone. The study revealed variation in microhardness profiles in both horizontal and vertical directions of the building primarily due to the consequence of varying heat input from successive additive layer welding. Lu et al. [26] have also used this technique for fabricating multi-layers of AA2050 aluminum by stacking multiple layers of cast AA2050 in between wrought AA2050. Studies on hardness and fracture behavior were conducted on the fabricated part. Various studies reveal that the FSAM technique can be successfully incorporated for manufacturing additively manufactured parts in a solid state. However, various issues, including the presence of non-uniform microstructures and mechanical qualities in the construct due to complicated material flow and thermal exposure, prevent this technique from being widely applied to other industrial production processes.

Even though it is worth noting that few published works have shown the feasibility of the FSAM technique, this is the first effort to demonstrate the practicality of this technique to manufacture an alternating gradient composite structure of aluminum alloys in the solid state. The primary objective of this investigation is to explore the feasibility of using the FSAM technique to fabricate a multi-layered alternating gradient composite structure of AA6061-T6 and AA7075-T6, which is a first of its kind. In this work, investigations were performed to check the microhardness of the final part along the build direction and in the horizontal direction in different layers. The tensile characteristics of the sliced samples obtained in the longitudinal direction from various layers were also studied. Characterization of samples extracted from various parts was done using a field emission scanning electron microscope fitted with energy-dispersive X-ray spectroscopy (FE-SEM/EDS). The findings of the microstructure characterization and mechanical property investigation conducted in this research are compared to base metals to provide a baseline comparison. For this study, AA6061-T6 and AA7075-T6 were chosen for fabricating the alternating gradient composite structure primarily because of their potential use as structural components in various applications such as automotive and aerospace industries. The automobile industry has dramatically boosted its use of AA6061 aluminum alloy in recent years, particularly in fabricating doors and boot lids of cars. This is primarily due to the material’s lightweight properties which result in reducing energy use and thus benefiting the surrounding ecosystem [27,28]. The proposed alternating gradient composite structure of AA6061 and AA7075 can be substituted with the existing AA6061 owing to its increased strength and hardness and concurrently maintaining its lightweight properties.

## 2. Materials and Methods

Base materials (BM) selected for this study were 3 mm thick AA6061-T6 and AA7075-T6 aluminum alloy plates of dimensions 200 × 150 × 3 mm. Table 1 lists the elements present in these aluminum alloys. In this investigation, the FSAM technique was carried out by stacking several AA6061-T6 and AA7075-T6 plates alternatively on top of one another and then friction stir lap welding each plate to form the alternating gradient composite structure. Once two plates were joined, the surface was ground flat by milling and meticulously cleaned using ethanol. The surface was milled to eliminate the flash generated during the process and to ensure that the surfaces of each layer are in direct touch with one another. This procedure was repeatedly carried out until a total height of 18 mm was achieved.

It is essential to consider that a multitude of research has been done on the FSW behavior of AA6061-T6 and AA7075-T6 aluminum alloys [29,30,31,32,33,34,35], each with its own set of suggested welding parameters. Even when working with identical materials, the welding parameters may dramatically differ depending on factors such as the weld orientation, plate thickness, and, most crucially, the tool’s geometry. Multiple studies, both experimental and numerical, have been conducted to determine the relative importance of various pin profiles [36,37,38]. The majority of studies revealed that a threaded conical pin is most beneficial as higher material velocity and proper material flow can be achieved. Therefore, an H13 tool having a threaded, tapered, and conical pin was utilized in this study. To attain a wider weld zone, wide enough to extract a tensile sample, the shoulder diameter of the tool was selected as 24 mm along with a pin root diameter of 8 mm, pin tip diameter of 6 mm, and pin height of 4 mm as shown in Figure 2. As AA6061 and AA7075 plates were arranged alternatively, one over the other, two separate tools of the exact dimensions were used. One tool was utilized when the AA6061-T6 Al alloy was positioned on top, while the other tool was used when the AA7075-T6 Al alloy was positioned on top. Since both the plates had different alloying elements, the usage of two different tools would prevent the unwanted deposition of elements from one plate to another during the joining process.

To guarantee that the tool pin pierced the bottom surface of the top plate into the subsequent layer, a plunge depth of 4.15 mm was provided. This also ensured the generation of maximum frictional heat and shoulder force on the plates. A tool tilt angle of 2⁰ was provided from the FSW machine’s vertical axis to avoid undue flash build-up from accumulating on the margins of the weld and to confine the materials under the shoulder. A constant vertical load of 20 kN was also provided to compact the materials properly and to avoid porosity in the final build. Above the specified plunge depth, the tool tilt angle and vertical load were kept constant during the FSAM process.

Moreover, to identify optimum welding parameters, the process of parametrization of the tool rotation speed and tool traverse speed was done separately when the AA6061 plate was on top and when the AA7075 plate was on top. During lower tool rotation speeds (<1200 for AA6061 and <1100 rpm for AA7075), defects such as wormholes were identified near the tool pin’s advancing side (AS), probably because the material did not have enough plastic flow in it. In contrast, at higher tool transverse speeds (<40 for AA6061 and <50 mm/min for AA7075), tunneling defects were produced in which the materials were not properly returned to the AS of the tool pin during the welding process. As a result, during the FSAM process, the optimal tool rotation speed was determined to be 1200 rpm when AA6061 was on top and 1100 rpm when AA7075 was on top. Similarly, the tool traverse speeds were selected as 40 and 50 mm/min, as represented in Table 2. During the FSAM process, these ideal welding parameters showed no signs of defects such as wormholes or tunnels. Most of the other welding parameters showed some defects in their macro or microstructure.

Specimens were then cross-sectioned out of the additively manufactured part in a direction perpendicular to the weld, as in Figure 3. It was then polished so that the microstructure could be characterized, and the welds could be examined for flaws and to evaluate the hardness. The polished samples were then etched using a Keller’s reagent (2.5 mL HNO_3_ + 1.5 mL HCl + 0.5 mL HF + 95 mL distilled water) for 15 s to obtain the macro- and microstructural features in various parts of the build. Macrostructure studies of the samples were performed using AM4115T Dino-Lite-Edge digital microscope and microstructure studies were performed using Olympus optical microscope. Field emission scanning electron microscopy (FE-SEM) combined with energy dispersive X-ray spectroscopy (EDX) was used for the microstructural characterization and study of secondary phase particles using a Thermo Fisher FEI-Quanta 250 FEG Field Emission Scanning Electron Microscope. An EDAX-TSL device connected to the FEI-Quanta FE-SEM was used to test electron backscatter diffraction (EBSD) for obtaining the grain size distribution. Mitutoyo-HB210 model Vickers microhardness tester with a load of 200 kgf and 15 s dwell time was used to evaluate the hardness of the FSAM build in both vertical and horizontal directions as per ASTM E384 standards. Three separate samples were utilized for this purpose. In the vertical direction, the hardness profiles were measured at intervals of 0.25 mm along the center in the build direction. In the horizontal direction, the hardness profiles were measured across the center of each layer.

To characterize the mechanical performance of the FSAM build, two sets of tensile samples were extracted from various layers in the stir zone of the build in the welding direction as per ASTM E8 standards, as shown in Figure 3. The loading axis of the specimens was aligned in a manner that was perpendicular to the direction of the build. Six tensile samples were extracted from each set using a wire-cut EDM machine to verify the tensile strength and ductility of various layers and were compared with the base metals. SEM analysis of fractography was then carried out on the fractured samples using Zeiss EVO 18 scanning electron microscope to study the initiation and propagation of cracks.

## 3. Results and Discussions

### 3.1. Formation Characteristics

Figure 4 represents the macro image of the cross-sectioned view of the FSAM build comprising seven layers of alternate AA6061 and AA7075. From this figure, the condition of interfacial bonding between the layers can be seen quite clearly. This is mainly due to the use of dissimilar metals in each layer in the FSAM build. As in FSW, various microstructural zones such as nugget zones (NZ), thermo-mechanically affected zones (TMAZ), and heat-affected zones (HAZ) are visible in the final FSAM build. To demonstrate the interfacial development in more detail, high-resolution images of selected portions of the section are represented in Figure 5a–h. Figure 5a–c represent the NZ in the upper, middle, and lower layers, Figure 5d–f represent the HAZ and TMAZ of various layers on the advancing side (AS) and Figure 5g–i represent the HAZ and TMAZ of various layers on the retreating side (RS). Onion ring patterns formed in the NZ are visible in all layers as a result of extrusion and upward material flow. This pattern is created when materials are transferred from the retreating side and flow in an upward route through the thread of the tool in the direction of the AS of the welded zone. The presence of AA6061 and AA7075 in the NZ is mostly responsible for the contrast that can be seen between the onion ring patterns. In the optical pictures, the presence of AA6061 is seen in the sections of the rings with lighter shading, while the presence of AA7075 is seen in the regions with darker shading.

From Figure 5a–i, it can be seen that certain visible defects, such as hooks and kiss bonding, are present in various layers of the FSAM part which is identical to the findings reported by Yu Chen et al. [39] and Wronska et al. [40]. Since FSAM consist of multiple layers formed by repeated FSLW technique, the intricate nature of the material flow during this process leads to several types of flaws. Hook formation is one of the most prevalent flaws detected at the interfaces of welded plates in FSLW which usually progress towards TMAZ or NZ. Hook, which in reality is a natural phenomenon that is commonly observed in dissimilar FSLW, can be described as unbonded interfaces formed due to the deformation of the faying surfaces which is observed on both AS and RS as observed in Figure 5d–f,g–i. There is a noticeable change in the hook’s direction of movement in AS and RS. The hook extends towards the NZ in the RS, whereas in AS it moves upwards towards the TMAZ and HAZ. Nevertheless, from Figure 5g–i it is evidently clear that the hooks are not stretching into the center of the NZ. High-resolution images reveal several overlapping transition zones (Figure 5b) beneath each interface, as well as a few kiss bonds (Figure 5f) nearer to the top edge. The primary reason for this is due to the intricate flow of materials and the formation of oxides during the FSAM process.

The creation of the hook in this current investigation is associated with the flow of material occurring in the NZ, which is influenced mainly by the pin having right-handed threads. The pin and the shoulder both play a vital role in driving the flow of the plastic material during FSLW. At the AS, the plastic materials move downwards along the thread and become deposited at the tip of the pin, thus forming a zone of concentrated material in the NZ. As welding progresses, the area of concentrated material becomes larger. Because of this, the plasticized materials that are located in the bottom plate, which is farther away from the tool pin, flow in the direction of the upper plate. Consequently, near the boundaries of the material-concentrated zone, the lap intersection bends upwards. The process described above is repeated, thus forming a hook-shaped structure which bends away from NZ and moves towards TMAZ in the AS and towards NZ in the RS (Figure 6a).

During the FSAM process, when another layer is added by FSLW, the newly added material moves downwards alongside the pin and it extrudes the hook, causing it to extend further. Sequentially, the hook becomes bent upwards in the direction of the TMAZ in the RS and further bends outwards in the AS (Figure 6b) as a consequence of the material flow in the upward direction, which is caused by the pin of the tool. During the dwell time, the tool shoulder penetrates the top surface of the upper layer and the tool pin penetrates the top surface of the lower layer. This action supplies the driving power required for the upward movement of the materials in the lower layer as shown in Figure 6b. Aydin et al. [41] and Ji et al. [42] reported similar outcomes. In addition, the creation of the kissing bond is due to the inadequate flow caused by the pin’s incomplete stirring. The development of the overlapping transition zone underneath the pin is related to the varying thermal cycles as a result of the pin’s multiple stirring actions as shown in Figure 6b.

### 3.2. Evolution of Microstructure

The material flow features in the FSAM components are particularly complicated since it involves numerous layers of friction stir lap joints, each of which is subject to variable degrees of temperature cycles. As a direct result of this, there are discernible distinctions in the microstructures of the various non-interfaces as well as the interfaces that exist between the layers. The initial microstructure of the aluminum base metals plays a substantial part in the development of the final microstructure of the FSAM build. Figure 7 represents the microstructure imageries of the base metals AA6061-T6 and AA7075-T6 which are distinguished by coarser and banded grains that are arranged in the direction of rolling.

The FSAM build is cross-sectioned from different areas to view the progression of the microstructure as shown in Figure 8. Following this, optical microscopy is utilized to scrutinize the microstructures of the sectioned areas at the various points as indicated in Figure 8. From the microstructure images (Figure 9a–i), it can be evidently seen that there are no defects such as micropores or voids in any of the areas. It is essential to take notice that none of the layers exhibited any volumetric faults that might be harmful to the mechanical characteristics. It can be particularly highlighted that no wormhole flaws were observed on any side of the NZ profile, suggesting that the parameters selected for this investigation supplied adequate energy to enable appropriate mixing of the material and adequate metallurgical bonding. Uniform mixing of dissimilar materials can also be observed from the micro images. During the FSAM process, the materials are subjected to varying thermal cycles, strain rates, and high peak temperatures resulting in dynamic recrystallization, coarsening, and dissolution which may all occur concurrently. The micro images of the NZ (Figure 9a–c) reveal fine grains that are equiaxed in nature which is due to dynamic recrystallization. The development of the microstructure in the NZ is a highly complicated process, which may be summarized as follows: (a) In the first stage, the grains undergo a process that results in the splitting of the initial grains forming a coarse banded structure. With the increase in strain, newer elongated grains that are fibrous in nature are generated, and subsequently, when additional grain subdivision happens continually, grains of finer sizes are developed. (b) As the temperature of the tool increases, very fine grains are produced from the high-angle grain boundaries that are closely spaced. Strain generated during the process causes the groups of fine grains to come closer together, which increases the volume fraction. (c) Lastly, the fragments of fibrous grains that are unstable create a microstructure that resembles a complete nugget which is composed of ultrafine grains with low aspect ratios. Static annealing, which occurs after welding, somewhat coarsens the grains and makes them more equiaxed.

Figure 9d–f reveal some refined variations in microstructures between the interfaces and non-interfaces alongside the direction of the build. Apparently, the grain size at distinct locations differs significantly from one another. From micro images, the grain sizes in TMAZ and HAZ, which are mainly affected by the shoulder, are fairly larger when compared with that in NZ. The primary cause of this phenomenon is the coarsening of the recrystallized fine grains beneath the shoulder area as a result of increased forging force, high peak temperatures, and extended cooling time. As can be observed in Figure 9e,f,h, the grain sizes are likewise variable, with smaller grains present in the noninterface region compared to that in the interface region. A newer transition region is also observed, which is characterized by the formation of banded structures (Figure 9e,i) having mostly coarser grains. Initial investigations indicate that when a second layer is being manufactured additively, the interface experiences repeated stirring action from the pin, which helps to refine the coarse grains present in the shoulder-driven zone. At the same time, owing to a longer length of thermal cycles, the interface grain sizes are larger than those in the noninterface. The use of dissimilar materials also aided in the formation of banded structures. During the FSAM process, the plastic movement of the materials and numerous repeated thermal cycles are thought to be responsible for the creation of the transition zone that is located near the interface. In addition, it is found that the grain size becomes bigger from the top of the build to the bottom as one move in the direction of the build thickness [43,44]. This is because, when multiple layers are being manufactured additively in the solid state, the bottom regions of the plates are exposed to a greater amount of thermal cycle thus resulting in a longer period of static annealing. Both these effects subject the recrystallized grains and precipitate particles to a more severe coarsening.

Figure 10a–h represents the inverse pole figures (IPF) corresponding to the various locations of the FSAM build obtained from EBSD analysis and Table 3 provides insightful information on the magnitude of fluctuation of the grain size at various locations. In the NZ largest average grain size of 3.3 μm was identified in the lowermost layer and the smallest average grain size of 1.7 μm was identified in the upper layer. A similar variation in grain size was also identified in TMAZ. As explained previously, the larger grain size is primarily due to inadequate strain rate and temperature.

### 3.3. Formation of Secondary Phases

The FE-SEM analysis was performed on the base metals and the final FSAM build, and the resulting micrographs are represented in Figure 11a–h, respectively. This was done so that the variations in the distribution of the secondary phase at each local point in the final build may be examined. When compared to the particles on the aluminum substrate in Figure 11a–b, the particles in the various places of the FSAM build had a more uniform distribution. After the FSAM process, the material underwent extreme plastic deformation and was heated to a high temperature. During this time, the coarse particles that reinforce the material were exposed to dissolving, breaking apart, and precipitate formation. In most instances, these procedures took place simultaneously, resulting in the formation of tiny precipitated particles which were distributed in a dispersive manner in the NZ as represented in Figure 11c–e. EDX microanalysis (Figure 12a–c) revealed that these precipitates may be made up of MgZn_2_ or AlZnMgCu formed from AA7075 and AlMg_2_Si formed from AA6061. The red circles in Figure 11c–h illustrates a significant size disparity between the particles found at the interfaces and those found in the noninterfaces. As observed in Figure 11f–h, the precipitated particles that form in the shoulder zone were much larger than those that were formed in the center NZ. When the metals were subjected to high temperatures and significant levels of stirring force, the combination of high strain rates and high temperatures may be sufficient to cause the particles of the secondary phase present in the aluminum base metals to dissolve. Additionally, the high temperatures generated, which were above the solvus temperature, in conjunction with a rapid rate of cooling helped the diffusion of grain boundaries to occur at lower temperatures, which ultimately led to the development of precipitates in the grain boundaries. On the contrary, the upper layer where the plastic flow of material was mostly dominated by the tool shoulder was exposed to higher frictional heat and extended cooling periods. This resulted in coarsening and aberrant growth of the fine precipitates during static annealing that followed. Moreover, from Figure 11e,f,h, banded precipitates can be observed near the interfaces. This is because the interfaces underwent lower strain rates and cooling rates than the individual layers.

### 3.4. Microhardness Evaluation

In the FSAM build, the change in hardness is defined mainly by the change in microstructure. Figure 13 depicts the microhardness profile along the horizontal direction in various interfaces of the stir zone and Figure 14 represents the profile of microhardness along the build direction vertically. The base metals had varying degrees of hardness, ranging from 119 to 121 HV for AA6061-T6 and 191 to 194 HV for AA7075-T6, respectively. According to the hardness results, the final FSAM build’s hardness gradually increased from the lower layer to the upper layer along the build direction. These results were comparable to those discovered in previous investigations that were of a similar type [19,23]. The maximum value of hardness was found to be 182.3 HV near the top layer and a minimum value of 72.6 HV was obtained at the bottom-most layer. As observed in Figure 13, the upper layer was noticeably having more hardness than the bottom. Figure 14 depicts a similar trend in hardness variation in the vertical direction, but it shows that the hardness drops somewhat in each transition zone that overlaps the previous one. These areas are highlighted using blue circles in Figure 14.

The strength and hardness of an FSAM structure are determined largely by the size, type, and amount of the strengthening precipitates and grains generated in the construction. From Figure 9a–i, very fine grains that are equiaxed were formed in the top layers and coarse grains were found in the bottom layers. Both AA6061-T6 and AA7075-T6 were precipitation-strengthened aluminum alloys. Hence, softening of materials in NZ was usually difficult in the FSW process. However, in the FSAM process, the NZ experienced high temperatures and strong plastic deformation resulting in fine grains with equiaxed boundaries. Therefore, in agreement with the Hall-Petch relationship [45], the hardness of the NZ was always greater than the Al base metals. The hardness was also enhanced by the FSAM process because the precipitate particles that were formed during this technique were smaller and more uniform in size. As seen in Figure 14, the component as a whole had an uneven distribution of its hardness along the direction of its thickness, and the hardness increased from bottom to top. The reduction in hardness in the bottom layer was a consequence of the multiple pass heat cycles that occur during the FSAM process which is equivalent to uninterrupted static annealing. This results in the creation of coarser grains of varying degrees in the bottom layers, causing a drop in hardness from the upper layer to the lower layer. Additionally, the same phenomena take place in the overlapping transition zones that are located close to the interfaces. As a consequence of the precipitate particles and grains being coarser as a result of the multiple stirring of the tool pin, there may be a minor decrease in the level of hardness experienced in each transition zone as represented in the circles in Figure 14. Therefore, to enhance the properties of the FSAM final build, it is of utmost necessity to manage the uniformity in microstructure and to increase the dissolution of precipitates in the build direction. This can be accomplished by the application of suitable heat inputs in conjunction with a controlled cooling process.

### 3.5. Tensile Properties

As the final goal of this FSAM process was to produce an alternating gradient composite structure of dissimilar aluminum-based components that was complete, free of defects, and had improved mechanical performances, an analysis of tensile properties was conducted on the various slices of the final FSAM build. The change in tensile curves of various distinct slices is shown in Figure 15a,b, depicting the values of the UTS and percentage of elongation of the various slices. The tensile characteristics of the various slices of the construction varied and agreed with the findings of microhardness characteristics. It can be observed that the UTS of various slices of the FSAM sections rises visibly when compared to the base material AA6061-T6, but was found to be lower than AA7075-T6. Simultaneously, the elongation of various slices was found to be less than that of the base metals. The uppermost slice had the greatest UTS, measuring 420 MPa, while the lowermost slice had the least UTS, measuring 316 MPa. In contrast, the ductility findings are the opposite, coming in at 10.4% at the uppermost slice and 14.8% at the lowermost slice.

During the FSAM process, the base metals underwent intense plastic deformation and were subjected to high temperatures, which resulted in the creation of small equiaxed grains and homogenous precipitate particles in the NZ as observed in Figure 9a–c and Figure 11c–e. This intense plastic deformation led to an increase in the concentration of internal dislocations and the elevated temperatures made it easier for the dislocations and other crystalline defects to interact with each other. These two factors led to an increase in dislocation motion resistance [43,46]. As a result, the presence of fine grains may extend the distance over which dislocations migrate at the time of tensile testing while simultaneously it reduces the formation of dislocation pileups. In addition, precipitate particles that were more uniform and dispersed acted as a hindrance to the mobility of dislocations. This resulted in a reduction in stress concentration and eliminated the formation of minute cracks in the grain boundaries. Even if minute cracks were generated, a greater resistance to crack propagation would increase the tensile strength of the final FSAM build. The reason for the reduction in UTS of the slices at the bottom layers was due to the fact that the precipitate particles and grains became coarsened dramatically as a result of incessant static annealing, which results in the formation of a significant amount of microcracks and stress concentration [47].

### 3.6. Fractography

Using SEM, the tensile specimen’s fractured surfaces, comprising various slices and the aluminum base metals, were characterized and are represented in Figure 16a–h. As represented in Figure 16g–h, the fracture surfaces of the aluminum base metals comprised a variety of different-sized and shaped dimples, and at the same time, a collection of coarser particles of the secondary phase that have an elongated form peeled away from the surface. Looking at the fracture surfaces of the slice samples (Figure 16a–f), it can be observed that the slices in the upper layer had deeper dimples in comparison to the bottom layers and the tearing edges were found to be thicker with plenty of micropores indicating that the top layers fail in a ductile manner. On the other hand, the fracture surfaces in the bottom layer exhibit lesser dimples and tearing ridges when compared with those of the top and middle layers. Additionally, the number of voids that vary in size and shape was significantly reduced. Based on these observations, it can be deduced that the failure exhibits a fracture pattern that is brittle in nature characterized by quasi-cleavage and river markings. The primary reason for this is that the coarse precipitates that exist along the grain boundaries are susceptible to crack initiation under the lower value of stresses during tensile deformation. The thick tear ridges allow the tensile samples that are ductile in nature to withstand the tensile loads, but the samples that were brittle in nature collapse as soon as the void coalescence began.

## 4. Conclusions

By using a novel technique of FSAM, it was possible to effectively construct a multilayered alternating gradient composite structure using AA6061-T6 and AA7075-T6 Al alloys with the strength and hardness greater than AA6061-T6 but less than that of AA7075-T6. Investigations were conducted on the final FSAM build’s mechanical capabilities, as well as its formation features and microstructures. The following inferences can be made from the findings that were achieved from various theoretical findings and experimental results:(1)Initially, in the course of the joining of two plates using the FSLW process, the hooks formed in the AS stretch outwards away from the NZ, while in RS it moved towards the NZ. As the next plate was joined in the FSAM process, the hooks on the RS tended to bend outwards away from the NZ, because fresh plasticized material was extruded from the top plate. As a result, a perfectly finished stack of seven layers of an alternating gradient composite structure comprising alternate AA6061-T6 and AA7075-T6 was manufactured;(2)In NZ, the dynamic recrystallization caused the grains to become fine and have almost the same dimensions in all directions. However, the size of the grains varied depending on where they were located. Static annealing caused the grains at the bottom layers of the sample to become coarser, while the grains at the top layers became finer. In addition, certain banded grains with a coarser texture were generated in the overlapping transition zones. The precipitate particles that are formed also exhibit similar trends in their distribution, size, and structure;(3)In comparison to aluminum base metals, the UTS of the FSAM build was higher than the tensile strength of AA6061-T6 but was lower when compared to AA7075-T6. In addition, there was a reduction in elongation of the FSAM build when compared with both the base metals. Moreover, it was noticed that the tensile strength of the various slices increased steadily along the direction of the build, reaching a maximum value of 420 Mpa in the topmost layer. In general, the final FSAM component exhibited a greater strength which is comparable with the hardness values;(4)The fractography studies reveal that the specimens from the upper layer have lots of dimples of varying sizes, forms, and thicker ripping ridges, while the bottom layers present a fracture that is brittle in nature brittle fracture having river patterns and quasi-cleavage patterns.

## Figures and Tables

**Figure 1 materials-15-07369-f001:**
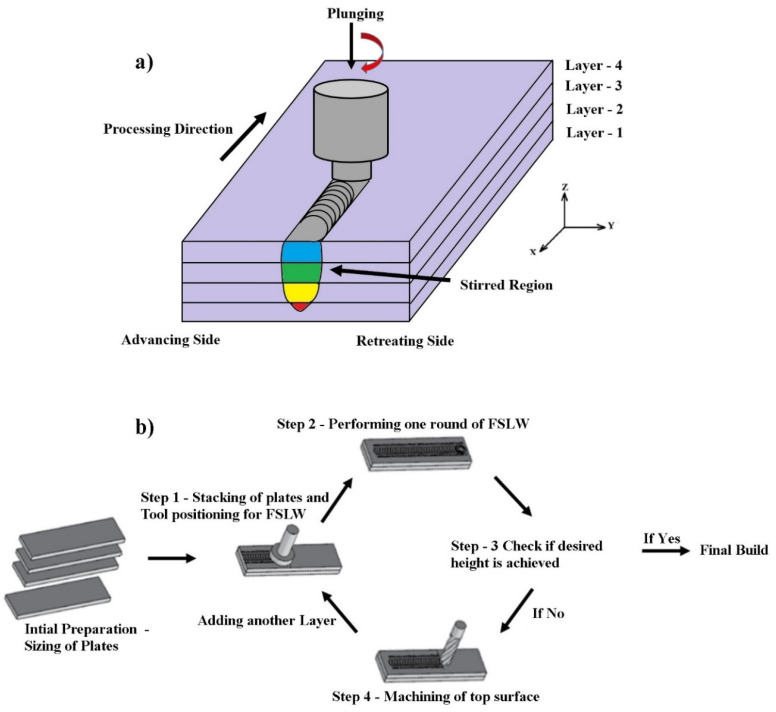
FSAM process: (**a**) Schematic representation of FSAM process and (**b**) steps in FSAM process.

**Figure 2 materials-15-07369-f002:**
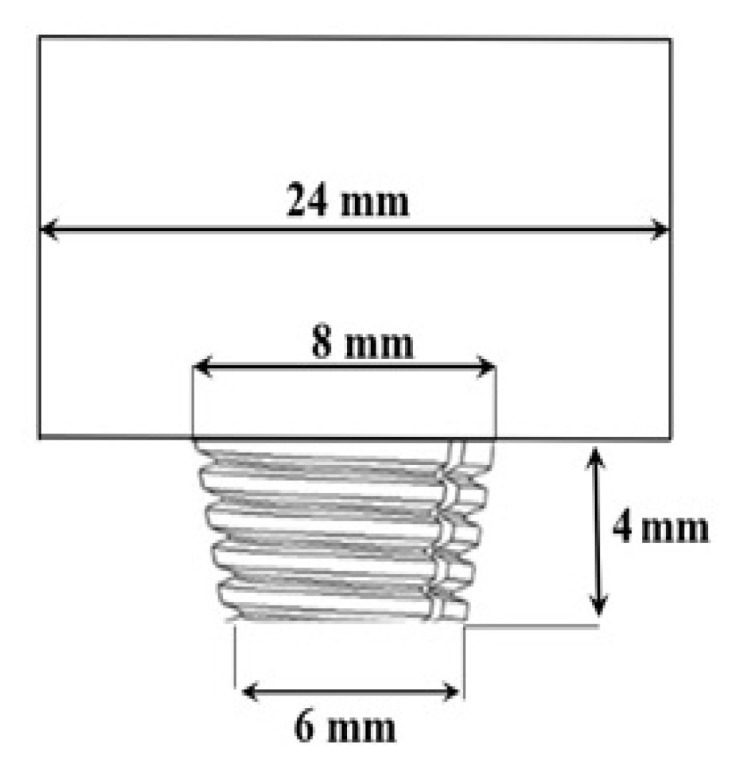
Schematic diagram of FSW tool used for FSAM.

**Figure 3 materials-15-07369-f003:**
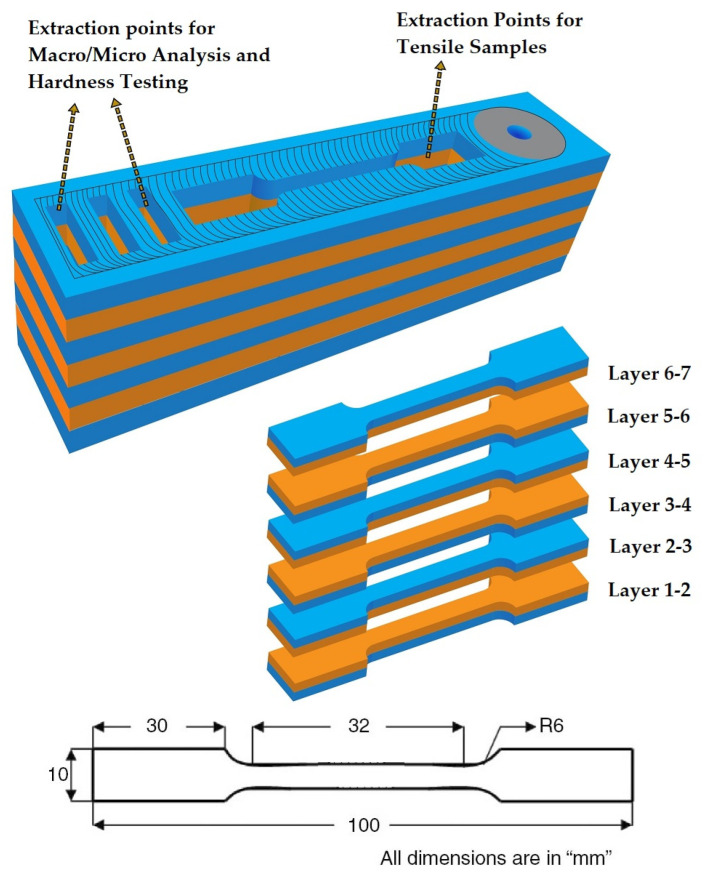
Diagrammatic representation of sample extraction points and tensile sample dimensions.

**Figure 4 materials-15-07369-f004:**
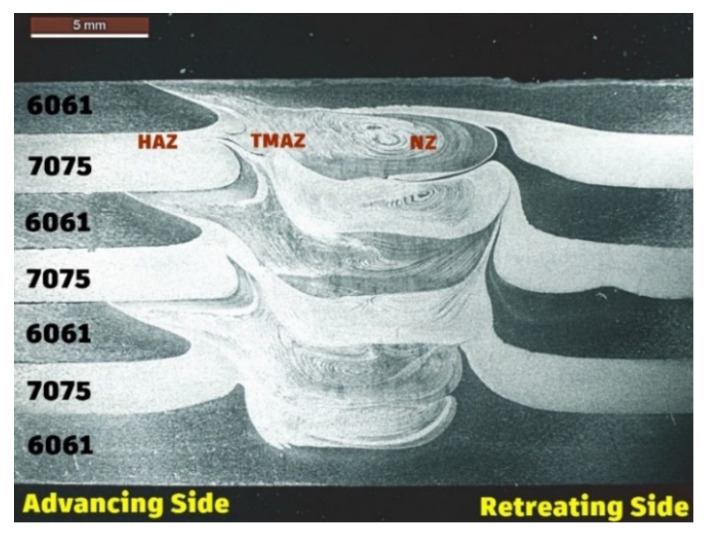
Cross-sectioned view of the FSAM build.

**Figure 5 materials-15-07369-f005:**
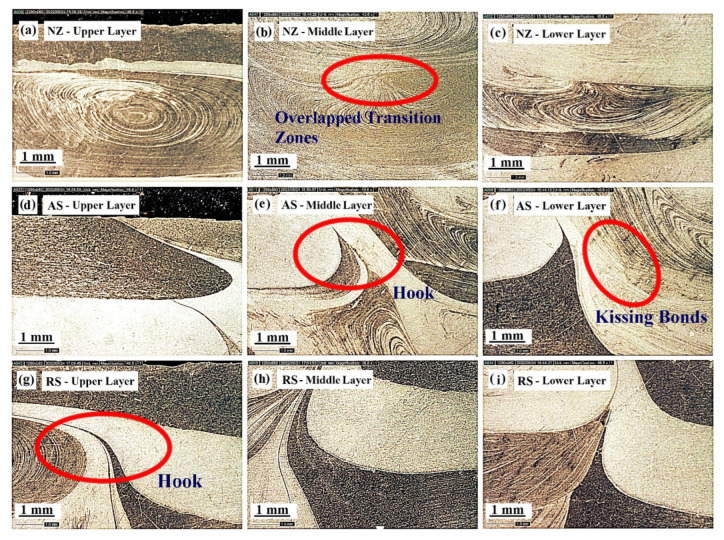
Formation morphology found in cross-section (**a**–**c**) NZ in upper, middle, and lower layer, (**d–f**) AS in upper, middle, and lower layer, and (**g**–**i**) RS in upper, middle, and lower layer.

**Figure 6 materials-15-07369-f006:**
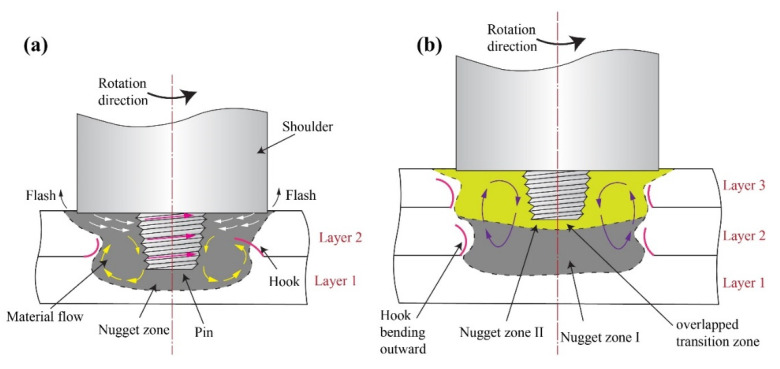
Schematic diagram of metal flow (**a**) with two plates and (**b**) with multiple plates.

**Figure 7 materials-15-07369-f007:**
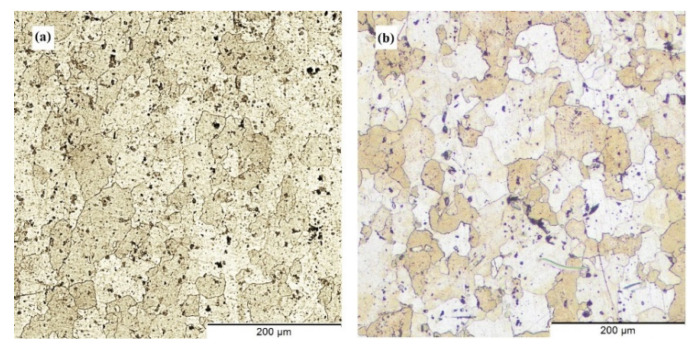
Microstructure of the base metals (**a**) AA6061-T6 and (**b**) AA7075-T6.

**Figure 8 materials-15-07369-f008:**
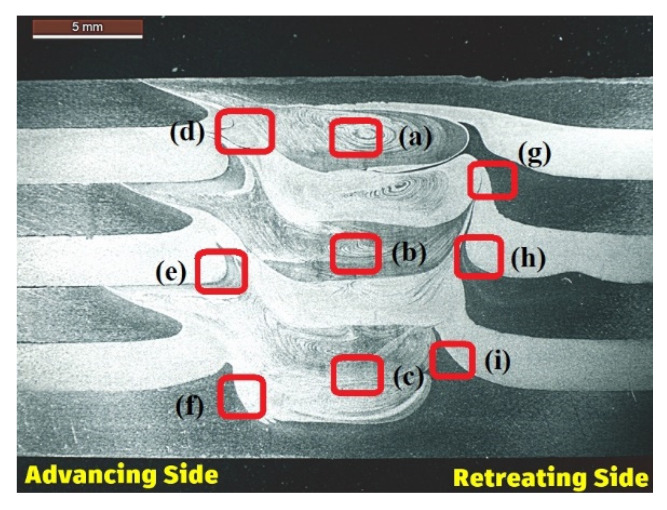
Extraction points for microstructural analysis. (**a**–**c**) NZ in upper, middle, and lower layer, (**d**–**f**) AS in upper, middle, and lower layer, and (**g**–**i**) RS in upper, middle, and lower layer.

**Figure 9 materials-15-07369-f009:**
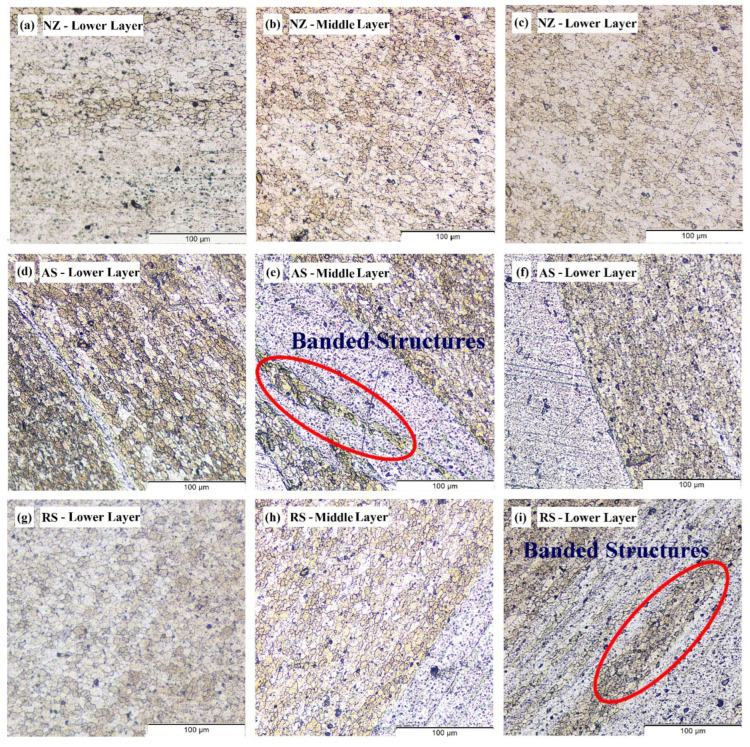
Microstructure of various areas in FSAM build (**a**–**c**) NZ in upper, middle, and lower layer, (**d–f**) AS in upper, middle, and lower layer, and (**g**–**i**) RS in upper, middle, and lower layer.

**Figure 10 materials-15-07369-f010:**
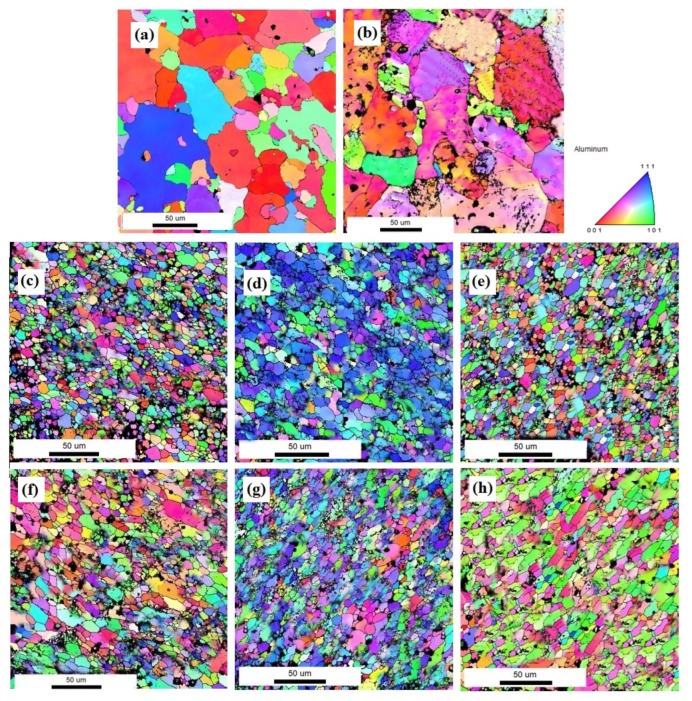
IPF maps portraying grain distribution in (**a**) BM-AA6061, (**b**) BM-AA7075, (**c**–**e**) NZ in upper, middle, and lower layer, (**f**) TMAZ of upper layer in AS, (**g**) TMAZ of middle layer in RS, and (**h**) TMAZ of lower layer in AS.

**Figure 11 materials-15-07369-f011:**
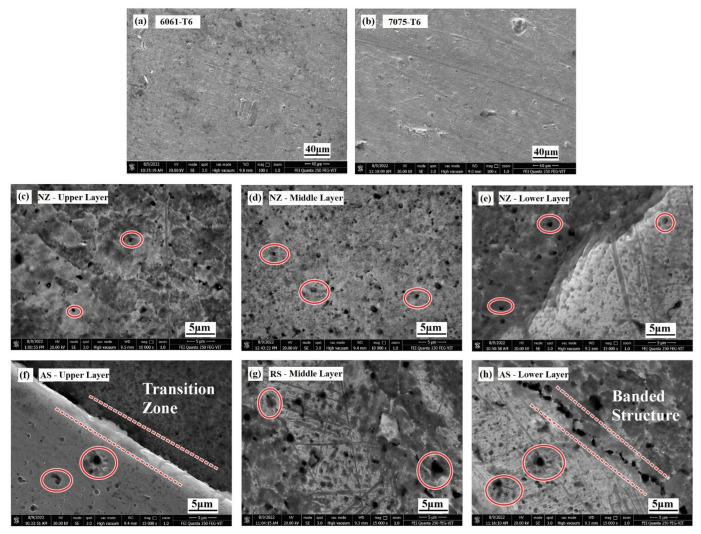
FE-SEM images of distribution of precipitates at various zones: (**a**,**b**) base metals AA6061-T6 and AA7075-T6, (**c**–**e**) NZ in upper, middle, and lower layer, (**f**) TMAZ of upper layer in AS, (**g**) TMAZ of middle layer in RS, and (**h**) TMAZ of lower layer in AS.

**Figure 12 materials-15-07369-f012:**
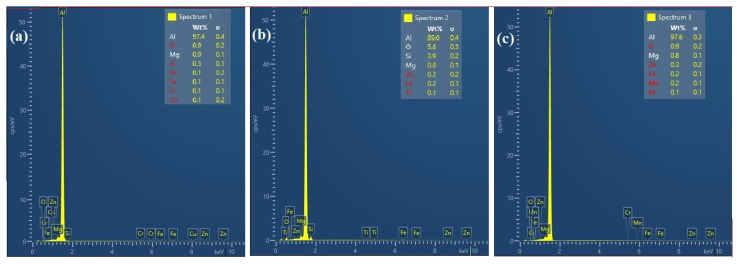
EDS analysis (**a**–**c**) NZ in upper, middle, and lower layers.

**Figure 13 materials-15-07369-f013:**
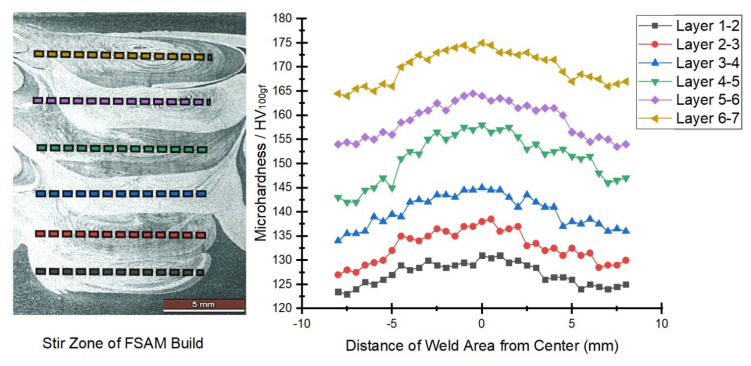
Microhardness analysis along horizontal direction.

**Figure 14 materials-15-07369-f014:**
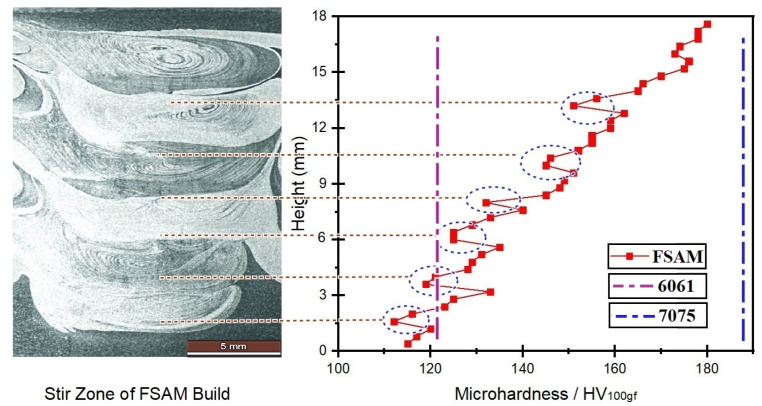
Microhardness analysis along vertical direction.

**Figure 15 materials-15-07369-f015:**
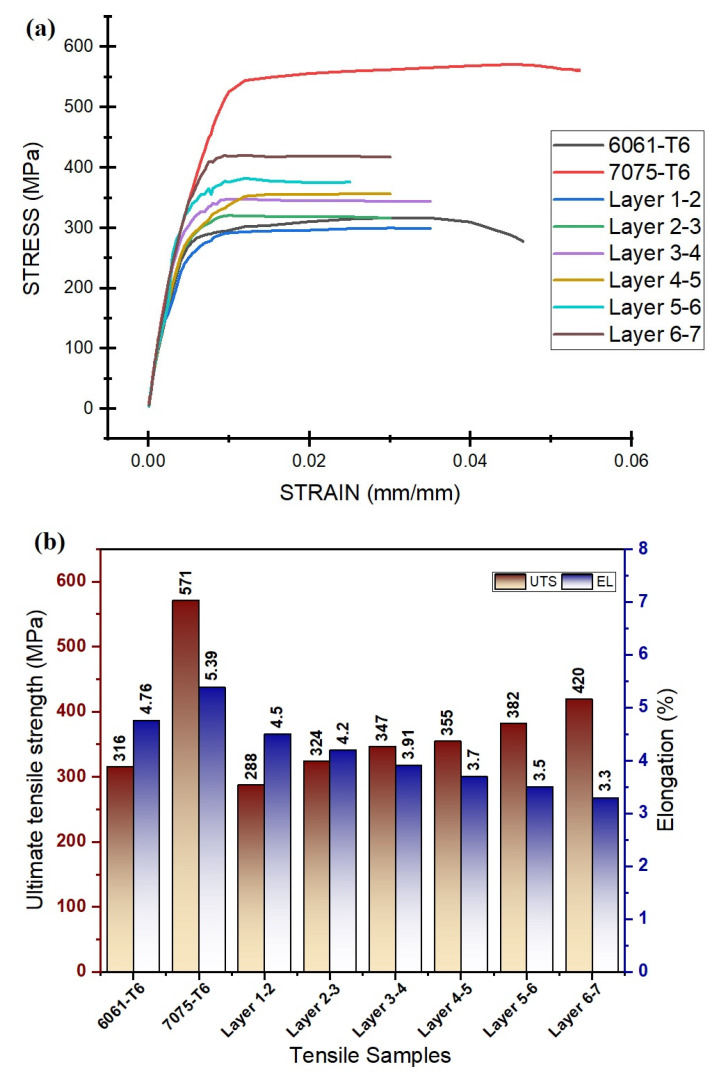
Tensile test results: (**a**) stress–strain curve; (**b**) UTS and elongation %.

**Figure 16 materials-15-07369-f016:**
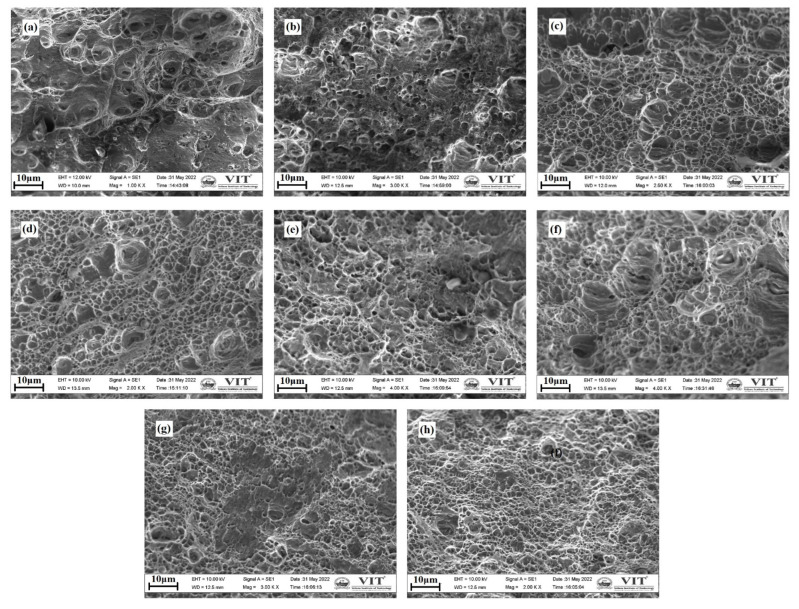
SEM micro images of fractured tensile samples: (**a**) layer 1–2, (**b**) layer 2–3, (**c**) layer 3–4, (**d**) layer 4–5, (**e**) layer 5–6, (**f**) layer 6–7, (**g**) AA6061, and (**h**) AA7075.

**Table 1 materials-15-07369-t001:** Elemental composition of studied AA6061-T6 and AA7075-T6.

AA6061-T6	Mg	Si	Fe	Cu	Zn	Cr	Mn	Al
	<0.5	0.35	0.44	0.18	0.15	0.11	0.099	Balance
AA7075-T6	Zn	Mg	Cu	Fe	Cr	Mn	Si	Al
	4.98	2.45	1.21	0.25	0.22	0.17	0.086	Balance

**Table 2 materials-15-07369-t002:** Tool parameters.

Sl No	Material on Top	Tool Rotation Speed (rpm)	Tool Traverse Speed (mm/min)	Tool Tilt Angle (Degrees)
1	AA6061-T6	1200	40	2
2	AA7075-T6	1100	50	2

**Table 3 materials-15-07369-t003:** Grain size distribution at various locations.

	Location	Average Grain Size (μm)	Maximum Grain Size (μm)	Minimum Grain Size (μm)
(a)	BM–AA6061-T6	21.3	54.6	7.2
(b)	BM–AA7075-T6	20.2	51.1	6.8
(c)	NZ of upper layer	1.7	10.5	0.19
(d)	NZ of middle layer	2.8	11.9	0.26
(e)	NZ of lower layer	3.3	12.8	0.32
(f)	TMAZ of upper layer in AS	9.7	24.1	0.79
(g)	TMAZ of middle layer in RS	10.2	24.7	0.82
(h)	TMAZ of lower layer in AS	12.6	36.3	1.1

## Data Availability

Data are contained within the article.

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
