# Peer review of "Novel Technique for Design and Manufacture of Alternating Gradient Composite Structure of Aluminum Alloys Using Solid State Additive Manufacturing Technique"

_materials, 2022, doi:10.3390/ma15207369_

Round 1

Reviewer 1 Report

In this work, the authors have developed alternating two Gradient Composite Structure of Aluminium Alloys using Solid State Additive Manufacturing Technique. They have employed friction stir additive manufacturing and investigated the mechanical properties.

1.       In the introduction section, more content is on Additive Manufacturing Technique. Please discuss the need to fabricate multiple layers of alternating gradient composite structure using alternate layers of AA6061-T6 and AA7075-T6 aluminium alloys. Mention  the applications where this 3mm composite is required in Industry.

2.        Why two gradient composite structure was chosen? In the literatures, authors have mentioned about 7 layer FGM. Give justification.

3.       H13 tool having a 172 threaded, tapered, conical pin was utilized in this study. Mention the reason for selecting this tool and its influence on crucial parameters.

4.       Though the Authors talk about dynamic recrystallization, grain size measurement are not given. It is required in different locations in Thermo Mechanically Affected Zone.

Author Response

Dear Reviewer,

The revised comments have been attached for your reference. Please refer to the attached file.

Reviewer 2 Report

This paper provided a friction stir additive manufacturing method to fabricate multiple layers of  composite structure using alternate layers of AA6061-T6 and AA7075-T6 aluminium alloys.  The research is interesting, but what is the potential application of such alternating plates? state this in the introduction part.

For mechanical performance, microhardness and tensile strenth, how many specimens do you use for each data point in Figure 12 and Figure 13? 

Author Response

(The authors gave the same response as above.)

Reviewer 3 Report

In the manuscript, the multilayered alternating gradient composite structure of 6061-T6 and 7075-T6 alloys was constructed by using a novel technique of FSAM. The formation features, microstructures and mechanical property of the final FSAM build were conducted.

1. Errors in the figure number and title number.

2. “As seen in Figure 11, the component as a whole has an uneven distribution of its hardness along the direction of its thickness, and the hardness increases from bottom to top.”: does the figure correspond to the analysis? 

3. “This can be accomplished by the application of suitable heat inputs in conjunction with a controlled cooling process.”: has the author conducted relevant experiments?

4. In this paper, there are many analyses of the particle size, and the following mechanical properties are all related to the particle size. Can the author quantify the dimensional data of the particle for comparative analysis?

Author Response

(The authors gave the same response as above.)

Round 2

Reviewer 1 Report

Authors have carried out revisions satisfactorily.

Author Response

Reviewer: 1

Comment: 1

Moderate English changes required.

Response:

Thank you for the careful and thorough reading of this manuscript. English language has been revised carefully as per the reviewer suggestions. The revised manuscript is attached herewith for your kind reference.

Reviewer 3 Report

Comment 1: corrections to language and text editing.

Comment 2: abstract should be refined.

Author Response

Reviewer: 3

Comment: 1

Corrections to language and text editing.

Response:

Thank you for the careful and thorough reading of this manuscript. English language has been revised carefully and few text editing also has been done were few mistakes were identified as per the reviewer suggestions. The revised manuscript is attached herewith for your kind reference.

Comment: 2

Abstract should be refined

Response:  

Thank you very much for allowing us to clarify the above query. We have refined the abstract as per the reviewers comeents without any loss of content. The revised manuscript is attached herewith for your kind reference.
